# Efficacy analysis of new copper complex for visible light (455, 530 nm) radical/cationic photopolymerization: The synergic effects and catalytic cycle

Jui-Teng Lin[1]*, Jacques Lalevee[2], Da-Chuan Cheng[3]*

1 New Photon Corp., New Taipei City, Taiwan, ROC, 2 CNRS, IS2M UMR 7361, Université de Haute-Alsace, Mulhouse, France, 3 Department of Biomedical Imaging and Radiological Science, China Medical University, Taichong, Taiwan, ROC

* jtlin55@gmail.com (JTL); dccheng@mail.cmu.edu.tw (DCC)

## Abstract

The kinetics and the conversion features of two 3-component systems (A/B/N), based on the proposed new kinetic schemes of Mokbel and Mau et al, in which a visible LED is used to excite a copper complex to its excited triplet state (G*). The coupling of G* with iodonium salt and ethyl 4-(dimethylamino)benzoate (EDB) produces both free radical polymerization (FRP) of acrylates and the free radical promoted cationic polymerization (CP) of epoxides using various new copper complex as the initiator. Higher FRP and CP conversion can be achieved by co-additive of [B] and N, via the dual function of (i) regeneration [A], and (ii) generation of extra radicals. The interpenetrated polymer network (IPN) capable of initiating both FRP and CP in a blend of TMPTA and EPOX. The synergic effects due to CP include: (i) CP can increase viscosity limiting the diffusional oxygen replenishment; (ii) the cation also acts as a diluting agent for the IPN network, and (iii) the exothermic property of the CP. The catalytic cycle, synergic effects, and the oxygen inhibition are theoretically confirmed to support the experimental hypothesis. The measured results of Mokbel and Mau et al are well analyzed and matching the predicted features of our modeling.

**Data Availability Statement:** All relevant data are within the manuscript.

**Funding:** This study was supported by the Agence Nationale de la Recherche (ANR agency) in the

## 1. Introduction

Light sources (lasers, LED or lamps) having light spectra ranging from UV (365 nm), visible (430 nm to 660 nm) to near infrared (750 nm to 950 nm) have been used for photopolymerization in both industrial and medical applications, such as dental curing, microlithography, stereolithography, microelectronics, holography, additive manufacturing, and 3D bioprinting [1–11]. Recently, copper complexes have been used as a new polymerization approach enabling the formation of acetylacetonate radicals by redox reaction to initiate the free radical polymerization (FRP) of acrylates or the free radical promoted cationic polymerization (CP) of epoxides [12–16]. The efficiency of copper complex (G1) based photoinitiating systems (G1/

form of the NoPerox grant. The China Medical University provided support in the form of a grant to DCC [CMU110-ASIA-11]. The New Photon Corp provided support in the form of a salary to JTL. The specific roles of these authors are articulated in the 'author contributions' section. The funders had no role in study design, data collection and analysis, decision to publish, or preparation of the manuscript.

**Competing interests:** The authors have read the journal's policy and have the following competing interests: Jui-Teng Lin is the CEO of New Photon Corp. This does not alter our adherence to PLOS ONE policies on sharing data and materials. There are no patents, products in development or marketed products associated with this research to declare.

iodonium salt (Iod)/*N*-vinylcarbazole (NVK) was investigated by Mokbel et al [16], using light source (LEDs at 375, 395, 405 nm).

However, in the G1 systems, most of the copper complexes have absorption peaks about 400 to 430 nm, which are still close to the ultraviolet spectrum having a small light penetration depth (few mm), comparing to that of visible light at about 500 nm (green) to 680 nm (red). A panchromatic light in visible (455 nm and 530 nm) were recently reported Mau et al [17] using two strategies, (i) modification of the electron donating substituent attached to the phenanthroline ligand; and (ii) introduction of a ferrocenyl group, in the bulky phosphorylated ligand, an iodonium salt (Iod) and ethyl 4-(dimethylamino)benzoate (EDB).

Mau et al [17] investigated and compared the the system G1/Iod/EDB with the new copper complexes (defined as G2). The photoredox catalytic cycle for the three-component system G2/Iod/EDB is summarized as follows. A visible LED was used to excite copper complex (G2) to its excited triplet state $G^*$, which interacts with Iod salt ($Ar_2I^+$) to produce oxodized-G2, G(II), and radical $Ar^o$. This radical couples with EDB to produce radical $EDB^o$, which further couples with G(II) producing radical EDB(+) and the regeneration of G2. Radicals $Ar^o$ and $EDB^o$ leads to FRP, whereas radical EDB(+) leads to CP in a system having monomers trimethylolpropane triacrylate (TMPTA) and (3,4-epoxycyclohexane)methyl-3,4–301 epoxycyclohexylcarboxylate (EPOX), for FRP and CP, respectively, in the so-called interpenetrated polymer network (IPN). Greater details will be shown later in Scheme 3 of the present article. Based on the measurements, Mau et al [17] have explored many new features and proposed or hypothesize mechanisms involved in the the dynamic profiles of the conversion efficacy. However, their phenomenological discussions are still lack of more precise conclusions which require a mathematical modeling as presented in this article.

As the theoretical-part of our previous experimental study of G1 and G2 systems, this article will present, for the first time, the kinetics and the conversion features of the 3-component system for the new copper complex G1 and G2 initiators, based on our previous G1 system by Mokbel et al [16] and the new scheme proposed by *Mau* et al [17] for both FRP and CP. The roles of co-additive including their dual functions of regeneration of initiator and generation of extra radicals for improved conversion. The present article will focus on developing analytic formulas for key factors influencing the conversion rates and efficacy for the interpenetrated polymer network (IPN) capable of initiating both FRP and CP in a blend of TMPTA and (EPOX.

The measured data of Mokbel et al [16] and *Mau* et al [17] and will be analyzed by our formulas, specially for the synergy effects (for improved FRP) from CP which could reduce the oxygen inhibition effects (on the free radicals) via the increase of oxygen viscosity. We will also analyze the role of the copper complex concentration, which is more sensitive for FRP than CP. We note that the scheme proposed in the present article is more general than the scheme proposed by Mau et al [17], which has ignored the bimolecular coupling term and concentrations of the initiators are not optimized. Furthermore, the general 3-component system (called as A/B/N), could include other components than the specific molecular groups G2/Iod/EDB of Mau at al [17], which is the special case of our scheme. For example, the A/B/N system used for the analysis of G1-system by Mokbel et al [16] could be revised to analyze another G2-system of Mau et al [17].

## 2. Methods and modeling systems

The kinetics and formulas for the conversion efficacy of two 3-initiator systems (A/B/N) will be presented, in which A is defined as G1 in system of Mokbel et al [16] and G2 in system of Mau et al [17]. We will present the Schemes (based on the experimental designs) to construct the associated kinetic equations, and the solutions of the conversion rate equations for the monomer

lead to the photopolymerization efficacy for both FRP and CP. Under certain limiting cases, analytic formulas are derived and used to analyze the measured and predicted features, without complex numerical simulations requiring all the rate constants such as kj and Kj, which are not yet completely available. However, our derived analytic formulas are able to provide enough details for the roles of each of the key parameters influencing the efficacy of FRP and CP.

## 2.1. Photochemical G1 system

As shown by Scheme 1, a 3-component system (A/B/N) defined by the ground state of initiator-A, which is excited to its first-excited state PI*, and a triplet excited state T having a quantum yield (q). The triplet state T interacts with initiator [B] to produce an oxodized-A (or [C]) and radical R, which interacts with co-initiator (or additive) N to produce radical S', which couples with [C] to produce a cation S and lead to the regeneration of [A]. Monomer M' and M coupled with radicals S' and S for FRP and CP conversion, respectively [18].

**Scheme 1**. The schematics of a 3-initiator system, (A/B/N), where A is the ground state of initiator-A, having an excited triplet state T, which interacts with co-initiator [B] to produce radical R and oxodized-A (or [C]); R interacts with co-initiator (or additive) N to produce radical S', which couples with [C] to produce a cation S and lead to the regeneration of [A]. Monomer M' and M coupled with radicals S' and S for FRP and CP conversion, respectively [18].

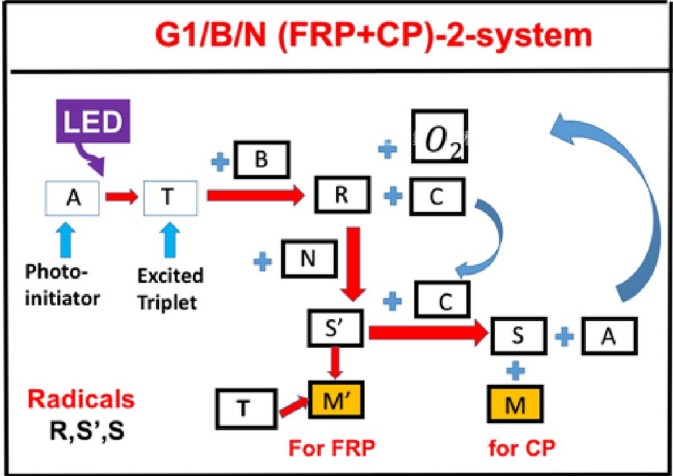

A specific measured system of related to Scheme 1 was reported by Mokbel et al [16], in which their Scheme 3 proposed the photoredox catalytic cycle for a 3-component (co-initiator) system of G1/Iod/NVK, where G1 is a copper complex in combination with iodonium salt (Iod), (oxidizing agent) generates the radical species through an electron transfer reaction. A propagation system containing the *N*-vinylcarbazole (NVK) additive leads to simultaneous regeneration of G1 and the formation of highly reactive cations (Ph- NVK+), which can very efficiently initiate the CP conversion.

The kinetic equations for our previous systems [21–23] are revised for the 3-initiator (A/B/N) and 2-monomer system (M'M), as follows.

$$\frac{\partial[A]}{\partial t} = -bI[A] + REG \tag{1}$$

$$\frac{\partial[B]}{\partial t} = -k_2 T[B] \tag{2}$$

$$\frac{\partial[C]}{\partial t} = k_2[B]\,T\, - \, k_4 S'[C] \tag{3}$$

$$\frac{\partial T}{\partial t} = bI[A]\, - \, (k_5 + k_2[B] + kM' + k_7[O_2])T \tag{4}$$

$$\frac{\partial R}{\partial t} = k_2[B]\,T\, - \, (k''[O_2] + k_6 N + k'R + k_8 S' + K''M')R \tag{5}$$

$$\frac{\partial N}{\partial t} = -k_6 RN \tag{6}$$

$$\frac{\partial S'}{\partial t} = k_6 RN - \, (k_4[C] + K'M')S' \tag{7}$$

$$\frac{\partial S}{\partial t} = k_4[C]S'\, - \, KSM \tag{8}$$

In Eq (1) REG = $(k_5+kM')T+k_4[C]S'+ k_7[O_2]T$ is the regeneration term of of the initiator, [A]. b = 83.6a'wq, where *w* is the light wavelength (in cm) and *q* is the triplet state T quantum yield; *a'* is the mole absorption coefficient, in (1/mM/%) and I (z, t) is the light intensity, in mW/cm$^2$. Based on Scheme 1 (which is supported by experimentally proposed scheme), construction of the kinetic Eqs (1) to (8) is straightforward by assigning coupling constants among the components as shown by our previous modelings [20–23]. For example, $k_j$ (with j = 1,2,3) are for the couplings of T and [A], [B], and [C], respectively; $k_4$ and $k_8$ are for the couplings of S' and [C], and R and M', respectively. The conversion due to coupling of T, and radicals R and S' with monomer M' (for FRP), and S with M (for CP) are given by the rate constants of $k_7$, $k_8$, K' and K, respectively. We have also include the oxygen inhibition effect [23] (for system in air) given by the k"R[O$_2$} in Eq (5). For system with laminate or when R is insensitive to oxygen (or k" is very small), oxygen inhibition is reduced and conversion is improved [22]. In above kinetics equations, we assume the bimolecular termination is mainly due to the coupling term R$^2$ shown in Eq (5), and ignore the weak coulings of S and S, S and S' and S'and S'.

The monomer conversions for FRP and CP are given by [21]

$$\frac{dM'}{dt} = -(kT + K''R + K'S')M' \tag{9}$$

$$\frac{dM}{dt} = -KSM \tag{10}$$

Above equations indicate that conversions for FRP and CP are given by the interaction of (T,R,S') and M', and S and M, respectively. We note that the co-initiator, [B] (or Iod) has dual function of enhancing FRP (via R and S') and CP (via S).

We note that by knowing the key coulings among various components in the 3-component (co-initiator) system one could easily construct the above Eqs (1) to (8) to a specific system. For example, Eqs (1) to (10) are constructed for the specific system of G1/Iod/NVK of Mokbel et al [16], using short hand notations: A = Cu(I); T = Cu*(I), B = Iod; N = NVK, C = Cu(II), R = Ar*, S' = Ar-NVK*, S = Ar-NVK$^{(+)}$, in system having two monomers M' = TAMPTA (for FRP conversion) and M = epoxy (for CP conversion), where Iod is iodonium salt, NKV is N-vinylcarbazole, and TAMPA is Trimethylol-propane triacrylate. They also compare the conversion of initiator A = cooper and A = bis(2,4,6-trimethylbenzoyl)-phenylphosphineoxide (BAPO).

## 2.2. Photochemical G2 system

Similar to Scheme 1 (for G1 system), Scheme 2 presents a more complex system to include the addition coupling of B (Iod) and N (EDB) which is used as a standard to ensure that the EDB/Iod charge transfer complex could not initiate the polymerization, if lack of absorption for the charge transfer complex at a non-absorbing light wavelength. It was also measured in ref [17].

As shown in Scheme 2, a 3-component system (A/B/N) defined by the ground state of initiator, [A], which is excited to its first-excited state PI*, and a triplet excited state T couples with an additive [B] to produce an oxidized-A (or [C]) and a radical R, which interacts with co-additive N to produce radical S'. Further coupling of S' and [C] produces cation S and leads to the regeneration of [A]. Monomer M' and M coupled with free radicals R and S' (for FRP) and cation S (for CP) conversion, respectively. We note that both Schemes 1 and 2 were also published in our recent Review article [18], which, however, only presented the efficacy key influencing features without any details of the kinetic equations or mathematical formulas as presented in this article.

**Scheme 2. The scheme chart of a 3-component system, (A/B/N), withe two monomers, M' and M, for the FRP and CP conversion, respectively, via radicals R, S' and S (see text for details)** [18].

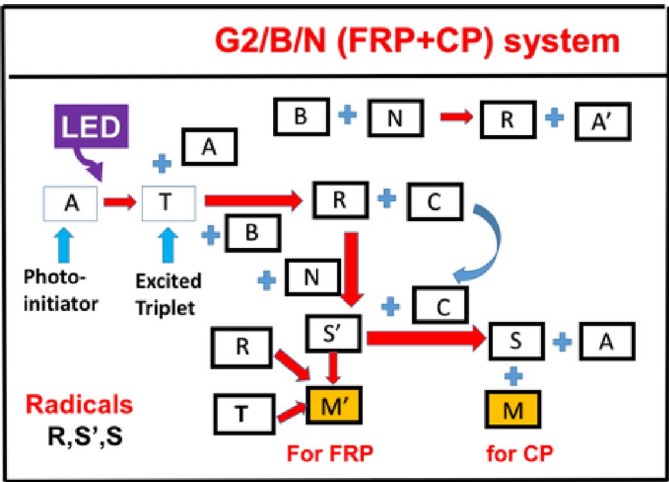

Specific measured system related to Scheme 2 was reported by Mau et al [17] proposed the photoredox catalytic cycle for a 3-component system of G2/Iod/EDB, where G2 is new copper complex, in combination with iodonium salt (Iod), (oxidizing agent) generates the radical species through an electron transfer reaction. The photoredox catalytic cycle for the three-component system G2/Iod/EDB is shown in Scheme 3 with the associated kinetic reactions shown in Scheme 4: (r1) for the light initiate copper complex (G1) to its excited triplet state $G^*$; (r2) coupling of $G^*$ with Iod salt ($Ar_2I^+$) to produce oxidized-G2, or G(II), and radical $Ar^o$, which, in (r3), couples with EDB to produce radical $EDB^o$ (r4), which further couples with G(II) producing cation EDB(+) and the regeneration of G2; Also shown is the oxygen inhibition effect (r5). The charge transfer between Iod and EDB (without the light excitation) produces radical $Ar^o$ and radical cation $EDB^o(+)$ (r6).

**Scheme 3**. Photoredox catalytic cycle for the three-component system G1/Iod/EDB—Adapted from Mau et al [17], where the G1 is redefined as G2 in the present article.

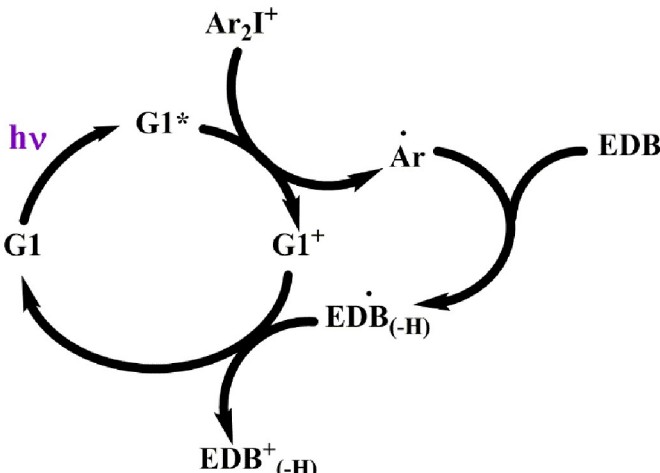

The two-component system Iod/EDB was also used as a reference, in which EDB can form a charge transfer complex with the iodonium salt (Iod) producing extra radical $Ar^0$ for FRP without the light as shown by (r6) of Scheme 4. The oxygen inhibition (OIH) is shown in (r5). We will also discuss the synergy effects (for improved FRP) from CP which could reduce IOH via the increase of oxygen viscosity. We note that both radicals of $Ar^o$ and $EDB^o$ lead to FRP (in monomer TMPTA), and has a higher conversion efficacy than that of CP (in monomer EPOX), which is produced only by the cation EDB(+). The formula showing higher FRP conversion efficacy will be derived later.

The kinetic Scheme 3 proposed by Mau et al [17] is translated into our Scheme 2 under the following short hand notations: A = Cu(I); T = $Cu^*$(I), B = Iod; N = EDB, C = Cu(II), R = $Ar^0$, S' = $EDB^o$, S = EDB(+), in system having two monomers M' = TAMPTA (for FRP conversion) and M = epoxy (for CP conversion). However, we note that Scheme 2 of the present article is more general than Scheme 3 of Mau et al [17], which has ignored the bimolecular coupling term R+R, the coupling terms of R and S, R and S', R and N. Therefore, the specific system proposed by the Scheme 3 of Mau at al [17] is a simplified case of our Scheme 2.

**Scheme 4**. The kinetic reactions for light initiated new copper complex (G2) for both FRP and CP conversation using free radicals of $Ar^o$ and $EDB^o$ (for FRP), and cation EDB(+) for CP. Also shown are the regeneration of G2 in (r4) and the the oxygen inhibition effect in (r5) and the charge transfer between $Ar_2I^+$ and EDB producing extra radical (R) (r6). See text for more details.

$$
\begin{aligned}
&\text{light} + \text{G2} = \text{G}^* && \text{(r1)}\\
&\text{G}^* + Ar_2I^+ = \text{G(II)} + Ar^\circ + ArI && \text{(r2)}\\
&Ar^\circ + \text{EDB} = \text{Ar-H} + EDB^\circ && \text{(r3)}\\
&EDB^\circ + \text{G(II)} = \text{EDB(+)} + \text{G2} && \text{(r4)}\\
&Ar^\circ + O_2 = \text{Ar-}O_2{}^\circ && \text{(r5)}\\
&Ar_2I^+ + \text{EDB} = Ar^\circ + EDB^\circ(+) + ArI && \text{(r6)}
\end{aligned}
$$

Based on Scheme 2 and 4, the kinetic equations for our previous systems [18–20] are revised for the 3-component system (A/B/N) with 2-monomer M' (for FRP) and M (for CP) as

follows.

$$\frac{d[A]}{dt} = -bI[A] + REG \qquad (11)$$

$$\frac{d[B]}{dt} = -(k_2 T + K_{12}[N])[B] \qquad (12)$$

$$\frac{dN}{dt} = -(k_6 R + K_{12}[B])N \qquad (13)$$

$$\frac{dT}{dt} = bI[A] - (k_5 + k_2[B] + kM')T \qquad (14)$$

$$\frac{d[C]}{dt} = k_2[B]T - k_4 S'[C] \qquad (15)$$

$$\frac{dR}{dt} = k_2[B]T + K_{12}[B]N - (k''[O_2] + k_6 N + k'R + k_8 S' + k_7 M')R \qquad (16)$$

$$\frac{dS'}{dt} = k_6 RN - (k_4[C] + K'M')S' \qquad (17)$$

$$\frac{dS}{dt} = k_4[C]S' - KSM \qquad (18)$$

In Eq (11) the regeneration (REG) term of of the initiator, [A] given by REG = $(k_5+kM')$T+ $k_4$[C]S'+ + $k_7$[O$_2$]T. b = 83.6$a'wq$, where $w$ is the light wavelength (in cm) and $q$ is the triplet state T quantum yield; $a'$ is the mole absorption coefficient, in (1/mM/%) and I (z, t) is the light intensity, in mW/cm$^2$. In Eqs (17) and (18), K$_{12}$ is the coupling rate constant between Iod and EDB, which also produces extra radical R (without the light), as shown by (r6).

All the rate constants are defined previously [21] and they are related by the coupling terms. For examples, k$_j$ (with j = 1,2) are for the couplings of T with [B] and [C], respectively; k$_4$ and k$_8$ are for the couplings of S' and [C], and R and M', respectively. The coupling of radicals R and S' with monomer M' (for FRP), and S with M (for CP) are given by the rate constants of K' and K, respectively. We have also include the oxygen inhibition (OIH) effect [22] (for system in air) given by the k"R[O$_2$} in Eq (15). For system with laminate or when R is insensitive to oxygen (or k" is very small), OIH effect is reduced and conversion is improved [23]. In above kinetics, we have also include the bimolecular termination term, k$_7$R$^2$ in Eq (6). Different from Eqs (1) to (8) for a G1 system, Eqs (11) to (18) includes the electron transfer between the amine and Iod, the K$_{12}$ N[B] term.

The monomer conversions for FRP and CP in G2 system are given by Eqs (9) and (10), similar to that of G1 system [19, 20].

## 3. Comprehensive formulas and discussions

### 3.1. Photochemical G1 system

For comprehensive modeling we will use the so-called quasi-steady state assumption [15, 18]. The life time of the singlet and triplet states of photosensitizer, the triplet state (T), and the radicals (R, S' and S), since they either decay or react with cellular matrix immediately after they

are created. Let $dT/dt = dR/dt = dS'/dt = dS/dt = 0$, which give the quasi-steady-state solutions: $T = bIg[A]$, $S' = k_6RN/[(k_4[C]+K'M')]$; $S = k_4[C]S'/(KM)$; and $g = 1/ (k_5 + k_2[B] + k_7[O_2] + K''M')$. However, the steady-state solution of R is much more complex (to be discussed later). The oxygen inhibition effect (OIE), included in g, reduces the free radical R, and hence the conversion of FRP if system is in air. The cation (S) and the convesion of CP, however, was not sensitive to oxygen. We note that under this quasi-steady state conditions, $k_4[C]S' = k_2[B]T$, and therefore RGE = bIA, which is a perfect catalytic cycle i.e., $d[A]/dt = 0$, with $[A] = A_0$ is a constant and enhances the conversion of FRP and CP, serving as a catalytic cycle. In general, [A] is a decreasing function of time given by $[A] = A_0 \exp(-Ft)$, with F being a depletion factor for the case of non-perfect cycle, and $F = 0$, for a perfect catalytic cycle.

Under the above quasi-steady-state solutions, we obtain the simplified equations as follows.

$$\frac{\partial[A]}{\partial t} = 0 \tag{19}$$

$$\frac{\partial[B]}{\partial t} = -k_2bIg[A][B] \tag{20}$$

$$\frac{\partial N}{\partial t} = -k_6RN \tag{21}$$

$$\frac{\partial R}{\partial t} = k_2[B]\ T - (k''[O_2] + k_6N + k'R + k_8S' + K''M')R \tag{22}$$

The dynamic light intensity is given by [21]

$$\frac{\partial I(z,t)}{\partial z} = -A'(z,t)I(z,t) \tag{23}$$

$$A'(z,t) = 2.3\big[(a' - b')[A] + b'[[A]_0 + q'\big] \tag{24}$$

where, a' and b' are the molar extinction coefficient (in 1/mM/%) of the initiator and the photolysis product, respectively; q' is the absorption coefficient of the monomer. Most previous modeling assumed a constant [A] in Eq (20) under a perfect cycle condition.

A full numerical simulation is required for the solutions of Eqs (11)–(18), which will be presented elsewhere. We will focus on simplified formulas for some limiting cases, such that many features and the enhancement effects related to the measured data of Mokbel et al [17] can be analyzed based on these analytic solutions.

The steady-state solution of Eq (22) for R, is much complex due to the bimolecular coupling $k'R^2$, and requires to solve Eq (25) as follows.

$$k'R^2 + GR - H = 0 \tag{25}$$

where $G = k''[O_2] + k_6N + k_8S' + K''M'$, $H = k_2[B]T$; with $T' = bIA_0$. Solving for R, we obtain

$$R = \left(\frac{1}{2k'}\right)(-G + \sqrt{G^{(2)} + 4k'H}) \tag{26}$$

We will consider the solution of R in two cases: case (i) for unimolecular termination dominant, or $G \gg k'H$, we obtain $R = (k_2T'[B]/G)\ (1-0.5H/G)$, which is an increasing function of H/G, or $(k_2T'[B]/G)$, for first-order with $0.5H \ll G$; and case (ii) for bimolecular termination dominant, with $H \gg GR$, we obtain, $R = [H/k']^{0.5}$, which is a nonlinear square root function of T'.

Using the steady state solutions of T, R, S and S', we may solve Eqs (9) and (10) analytically, but only under the condition of g = 1/(K"M') and G = K"M' and for a perfect cycle case with [A] = $A_0$, T' = $bIA_0$.

For case (i), unimolecular termination, R = ($k_2$T'[B]/(K"M'), and S' = $k_6RN$/[(K'M'), Eq (9) becomes,

$$\frac{dM'}{dt} = -kT' - [1 + k_6N/(K''M')] k_2[B]T' \tag{27}$$

which also requires solution of N and [B] from Eqs (12) and (13). We obtain first-order [B] = $B_0$ exp(-dt), with d = $k_2$T'/(K"$M_0$), using an approximated M(t) = $M_0$', and assume $M_0$' = $M_0$. Similarly, solving Eq (13), N(t) = $N_0$ exp(-d't), with d' = $k_6$d. Therefore, the time integral of Eq (27) gives us the conversion efficacy of FRP defined as CE' = 1-M'/$M_0$', normalized by $M_0$'.

$$CE' = kT'[t + k_2B_0H(t) + QN_0B_0H'(t)]/M_0 \tag{28}$$

where H(t) = [1-exp(-dt)]/d; H'(t) = [1-exp(-d"t)]/d", with d" = d+d'; and Q = $k_6$/(K"$M_0$). We note that both H(t) and H'(t) have a transient state proportional to t, and steady state of (kT'H) and (kT'H') are independent to the light intensity, noting that d = $k_2$T'/(K"$M_0$).

For case (ii), bimolecular termination R = [$k_2$T'[B]/k']$^{0.5}$, Eq (10) becomes,

$$\frac{dM'}{dt} = -kT' - [1 + k_6N/(K''M')] \sqrt{k_2[B]T'/k'} \tag{29}$$

Therefore, the time integral of Eq (29) gives us

$$CE' = kT't/M_0 + (2/M_0)[P(t) + Q'N_0P'(t)] \sqrt{k_2B_0T'/k'} \tag{30}$$

where P(t) = [1-exp(-0.5dt)]/d; P'(t) = [1-exp(-0.5d"t)]/d"; and Q' = $k_6$/(K"$M_0$). We note that P(t) and P'(t) also have a transient state proportional to t, same as that of H and H'. However, they have a completely different feature for their steady state. Both P and P' is proportional to (1/T')$^{0.5}$, or [1/($bIA_0$)]$^{0.5}$, which leads to a unique feature that higher light intensity has a lower steady state value than that of lower intensity. This feature was first discovered by Lin et al [21] in 2017, for the corneal crosslinking system, numerically and analytically.

For steady state value S = $k_2$[B]T/(KM), which allows us to calculate Eq (10), we obtain the CE for CP is given by the time integral of $k_2$[B]T,

$$CE = k_2T'B_0[1 - exp(-dt)]/(dM_0) \tag{31}$$

which has a transient state CE = $k_2A_0B_0$(bIt), and steady state CE = K"$B_0$, which is independent to the light intensity. The above formulas are based on the first order solution of g = (1/(K'M') and G = K'M', with M(t) = $M_0$. The second-order solution may use M(t) = M0—H"(t), with H"(t) being the integral of Eqs (9) and (10). However, most of the enhanced features of the CE of FRP and CP are shown in the first order formulas of Eqs (30) and (31), which will be used to analyze the measured data of Mokbel et al [17].

The dark reaction in CP is defined by the time the light is turned off (at t = ti), and the concentration of the initiator [B(t)], or Q(t)-function, remains virtually unchanged in time, i.e., Q(t2265ti) = constant, and the CE conversion for dark polymerization is given by Eq (31), but with t replaced by t-ti, in the factor exp(-t).

As shown by Eqs (9) and (10), and the approximated solutions of Eqs (28) to (31), the following significant features of the G1 system are summarized.

1. Co-initiator [B] has multiple functions of: (i) regeneration of initiator [A] leading to higher FRP and CP conversion; (ii) generation of radical S' for CP conversion, both are via $k_6bIg$ [A][B]. The regeneration term given by REG = $k_4$ [C]S' = bIg[A][B], with a REG factor F' = $1-(k_6/k_2)$ in Eq (26). For strong regeneration, (or when $k_6$ comparable to $k_2$), the reduction factor F' = $1-(k_6/k_2)$ reduces the depletion of initiator [A], and improves the CP conversion, Eq (26). For the extreme case of F' = 0, depletion of [A] due to light is totally compensated by the REG term, d[A]/dt = 0, and [A] = $[A]_0$, shown by Eq (21).

2. Co-initiator [N] has functions of: (i) generation of S' for FRP; (ii) generation of cation S for CP conversion; via $k_6R[N]$. We note that [N] always enhances steady-state FRP, via $QN_0B_0H'$ term in Eq (28), or $Q'N_0P'(t)$ term in Eq (30). However, steady-state CP conversions is independent to [N], given by Eq (31).

3. For unimolecular termination, the CE have a transient state proportional to t, and steady state independent to the light intensity, In contrast, for bimolecular termination, higher light intensity has a lower steady state value than that of lower intensity.

4. Dark polymerization is given by Eq (31), which provides the CP conversion even after the light is turned off. The lack of a termination mechanism for the CP conversion via the cationic intermediates which enables the polymerization to continue in living mode without requiring a constant input of light for propagation, thus offers an extended dark-cure reactions. Such kinetic behavior contrasts with that of the radical-mediated pathway of FRP, where radical and monomer reactions are almost immediately interrupted due to the effective exhaustion of the reactive radical intermediates. This dark polymerization also exists in thiol–Michael additions [13], but not in thiol–ene additions [14].

5. The oxygen inhibition effect ($k''[O_2]$), included in S' = ($k_6bIg[A][B]—k''[O_2]$)/[($k_4$ [C]+K'M')], which reduces the free radicals S', and thus the FRP conversion is lower in air comparing to in laminate [23]. However, the CP shown by Eq (31) indicates that CP is not sensitive to oxygen.

6. For thick polymers, the light intensity and the initiators concentrations are decreasing function of the depth (z), as shown by G(z) = $(1-k_1/(k_2[B]_0)) exp(-A'_0z)$. However, light intensity is an increasing function of time (t) due to the depletion of [A], unless for the extreme case that F' = 0 (a total compensation). Detailed temporal and spatial profile of conversion function require extensive numerical simulation which was published elsewhere [21, 24].

7. As reported by van der Laan et al. [7], photoinhibitor in a two-color system is strongly monomer-dependent, which also requires: (i) a high conversion of blue-photoinitiation in the absence of the UV-active inhibitor; (ii) a strong chain termination with significant reduction of blue and UV conversion in the presence of UV-active inhibitor and (iii) short induction time or rapid elimination of the inhibitor species in the dark (or absence of UV-light). Conversion efficiency may be also improved by reduction of the oxygen inhibition effect [9, 10, 12]. Synergic effects have been reported using co-initiator and/or additives [24–28], and 2- and 3-wavelength systems [29–31]. The present article presents the kinetics analysis of the co-initiator enhanced (catalyzed) conversion in FRP and CP which was reported by Mokbel et al [16], Garra et al [19] and Noribnet et al [20] in the 3-compnent G1/Iod/NVK system.

## 3.2. Analysis of maedured data (for G1 system)

Besides the general features described in section 3.1, our analytical formulas may be also used to analyze the measured results of Mokbel et al [16], in which quantitatively and precise comparison require the numerical simulatioms of Eqs (1) to (10).

1. Fig. 2 and Fig. 5 of Mokbel et al [16] for CP profiles of various epoxy functions, showing that G1/Iod/NVK has the faster raising rate and higher steady-state value than that of BAPO/Iod/NVK having a lower light absorption. This feature is shown by Eq (31), in which the conversion and rate function is proportional to the factor, $bI_0A_0(1+B_0)]$, which is an increasing function of the light absorption coefficient and the co-initiator concentrations $A_0$ and $B_0$; where $b = 83.6a'wq$, where $a'$ is the molar extinction coefficient, $w$ is the UV light wavelength and $q$ is the triplet state quantum yield. Given the values of $bI_0$ and the initial concentrations of the initiators, we can calculate and compare the profiles of CP in G1/Iod/NVK and BAPO/Iod/NVK systems which also need to know the rate constants such as kj, Kij, not yet available. Our formula, Eq (31), however, could predict the relative raising rates for various systems explored by Mokbel et al [16].

2. Fig. 3 of *Mokbel* et al [16] showed that higher dark polymerization in G1/Iod/NVK than BAPO/Iod/NVK. This feature may be easily seen by our Eq (31), which is increasing function of the the molar extinction coefficient. The dark polymerization exists in the free radical promoted CP and in Thiol–Michael addition polymerization [13], but does not exist in one-initiator FRP systems.

3. Fig. 4 and Fig. 7 of Mokbel et al [16] showed the effects of G1 concentration and sample thickness (24 um and 1.4 mm). In the case of thick samples (1.4mm) (Figure 4B), the CP conversion in G1/Iod/NVK system increases when decreasing the photoinitiator concentration, in contract to that of thin sample that higher concentration has higher conversion. These features may be analyzed by our thick polymer formula: $I(z) = I_0 \exp(-A''z)$, with $A'' = 1.15(a'+b')[A]_0$, which shows that increasing the concentration, $[A]_0$, (or larger $A''$) for thick sample (with z = 1.4 mm), the penetration of the light decreases, as shown by Eqs (23) and (24). For very thin sample (with z = 25 um), $A''z = 0$, and $I(z) = I_0$, independent to $[A]_0$. Therefore, there is an optimal initiator concentration in thick samples. We have previously [21] demonstrated mathematically the optimal $[A]_0$ value given by dG/dz = 0, with $G = [A]_0 \exp(-bzI_0[A]_0)$, to obtain the optimal concentration $[A]^* = 1/(bz I_0)$ which is inverse proportional the product of light intensity($I_0$), absorption coefficient (b) and the sample thickness (z). In contrast, for very thin sample, the conversion is always higher for higher $[A]_0$ and/or $[B]_0$ and there is no optimal values. The new finding of optimal initiator concentration in thick samples, demonstrated mathematically, requires further experimental investigations.

4. Fig. 12 and 14 of Mokbel et al [16] showed the oxygen inhibition effects (OIE) for system in air and in laminate. They showed that the FRP conversion of TMPTA was higher in laminate than in air. In contrast, the CP conversion of epoxy function was lower in laminate than in air. It may be because the FRP of TMPTA was faster than the CP, and most of the free radicals were consumed to initiate FRP. We also note that the radical for CP is much less sensitive to OIE than that of FRP, as also predicted by our formula, Eq (31). The participation of different thermal effects between thin and thick samples can also participate to some extend to explain the difference of behavior between CP and FRP. Mathematically, this features are shown by our formula, Eq (30). for FRP free radical (R) which is a decreasing function of oxygen, R = H/G, in which $k''[O_2] = 0$, in laminate. Our formulas, Eqs (30) and (31), also demonstrate that thick sample has less oxygen supply (inside the sample), such that OIE is smaller than that of thin sample. However, the FRP "volume" conversion of thick sample is still lower than thin sample due to the stronger light absorption loss in thick sample (the Beer-Lambert law), as shown by Fig. 14 (A) and (B) of Mokbel et al [16].

5. Fig. 14 of Mokbel et al [16] also showed that the addition of the Boltorn H2004 resin in the UviCure S105/TMPTA blend improves the final epoxy function. They explained as the decrease of cross-link density leading to a higher mobility of the reactive species. Lin at al [23] has developed a modeling for the role of oxygen inhibition and viscosity, in which lower viscosity (or higher mobility) leads to higher conversions.

### 3.3. Photochemical G2 system

Similar to the G1 system of 3.1, the quasi-steady state solutions for the G2 system: T = bIg[A], S' = $(k_6RN + k_2T[B])/(K'M')$; S = $k_2T[B]/(KM)$; and g = $1/(k_5 + k_2[B] + kM')$. The oxygen inhibition effect, included in g' and S', reduces the free radicals, R and S', and hence the conversion of FRP and CP if system is in air. We note that under this quasi-steady state conditions, $k_4[C]$ S' = $k_2[B]T$, and therefore RGE = bIA, which is a perfect catalytic cycle i.e., d[A]/dt = 0, with [A] is a constant. We have previously limited our formulas to the unimolecular coupling ok k'R [17]. However, the steady-state solution of Eq (6) for R, is much complex due to the bimolecular coupling $kR^2$, and requires to solve Eq (16) as follows [21, 22].

$$k'R^2 + GR - H = 0 \tag{32}$$

where G = $k''[O_2] + k_6N + k_7M'$, H = $k_2[B]T + K_{12}N[B]$; with T = bIg[A]. Solving for R, we obtain

$$R = \left(\frac{1}{2k'}\right)(-G + \sqrt{G^2 + 4k'H}) \tag{33}$$

Case (i) for unimolecular termination dominant, or G>>k'H, we obtain R = $k_2bIg([A][B]/G)(1-0.5H/G)$, which is an increasing function of H/G, or bIg[A][B]/G, for first-order with 0.5H<<G.

Case (ii) for bimolecular termination dominant, with H>> GR, we obtain, R = $[H/k']^{0.5}$, from Eq (32), or more precisely, from Eq (33),

$$R = \sqrt{H[1 + G^2/(8H^{0.5})]/k'} - G/(2k') \tag{34}$$

We note that in both cases, R is a decreasing function of the oxygen inhibition effect (OIH), the $k_5[O_2]$ in G. The OIH can be reduced via various strategies such as, Lin et al [21, 22]: (i) using a pre-irradiation of a red-light to eliminate the oxygen; (ii) an additive which could convert the unstable oxidized molecule produced from the coupling of R and oxygen, to radicals; (iii) the synergy effect of FRP and CP in an interpenetrated polymer network (IPN) system which to be detailed more later.

Under the above quasi-steady-state solutions, we obtain the simplified equations as follows for the G2-system, comparing to Eqs (19) to (22) for G1-system,

$$\frac{d[A]}{dt} = 0 \tag{35}$$

$$\frac{d[B]}{dt} = -(k_2bIg[A] + K_{12}N)[B] \tag{36}$$

$$\frac{dN}{dt} = -(k_6R + K_{12}[B])N \tag{37}$$

where we have used the steady state condition, $k_4[C]S' = k_2[B]T$, and REG = $(k_5+k'M')T+k_4[C]S'$, reduces to REG = $(k_5+kM'+k_2[B])T$, we found that Ib[A]-RGE = 0, a prefect catalytic cycle is

available, such that [A] is a constant., [A] = $A_0$, We note that this REG enhances the conversion of FRP and CP, serving as a catalytic cycle.

The equation for the FRP and CP conversion rate functions are given as follows for two special cases.

For case (i) unimolecular dominant, R = H/G = $k_2[B]T/G = (k_2 T+K_{12} [B])N/(k'M')$, with T = bIg[A].

$$\frac{dM'}{dt} = -kbIg[A]M' - 2k_2bIg[A][B] - (K_{12}[B] + k_6R)N \tag{38}$$

$$\frac{dM}{dt} = (K_{12}N + k_2bIg[A])[B] \tag{39}$$

For case (ii) bimolecular dominant, $R = [H/k']^{0.5}$, we obtain

$$\frac{dM'}{dt} = -- kbIg[A]M' - 2k'M'\sqrt{k_2bIg[A][B]/k'} - (K_{12}[B] + k_6R)N \tag{40}$$

$$\frac{dM}{dt} = - (K_{12}N + k_2bIg[A])[B] \tag{41}$$

The solutions of above equations lead to the conversion efficacy (CE) defined by CE' = 1- $M'/M'_0$ (for FRP), and CE = 1- $M/M_0$ (for CP). For analytic formulas, we will consider special cases of: (i) Iod/EDB system without light; (ii) G2/Iod system, with N = 0; (iii) G2/Iod/EDB systems, and under the steady state condition which leads to a constant [A] = $A_0$, for a strong RGE.

Case (A). Without the light, the CE due to the reaction between EDB and Iod is given by, when bI = 0, R = T = 0, with steady state radicals given by S' = $K_{12}[B]N/(K'M')$; S = $(K_{12}[B]N/(KM)$, solution of d[B]/dt = dN/dt = $-K_{12}[B]N$ is N = B+d, with d given by the initial conditions of [B] and [N]; [B] may be found by approximately as: [B] = N-d = $[Q—tK_{12}]$ = $Q(1- K_{12}Qt)$, with Q = $B_0(1+ 0.5d/B_0)$, which is a decreasing function of time (t).

Therefore Eqs (40) and (41) become

$$\frac{dM'}{dt} = \frac{dM}{dt} = -K_{12}[B]N \tag{42}$$

Time integral of [B]N, we obtain M(t) = $M_0—P(t)$, with P(t) = $Q(Q+d)t—0.5(2Q+d)Q't^2$ + $0.33Q'^2t^3$, with Q' = $K_{12} Q$, and Q = $K_{12}B_0(1+ 0.5d/B_0)$, which is a nonlinear increasing function of time (t), and proportional to $K_{12} B_0 N_0$. Therefore, the CE for FRP and CE' for CP are given by CE = 1- $M'/M'_0$, CE = 1- $M/M_0$, and both equals to P(t).

Case (B) for G2/Iod system, with N = 0. With [A] = $A_0$, the first-order solutions of Eq (2) is given by, [B] = $B_0$ -IbtA$_0$, for g = 1/ ($k_5+ k_2[B]+k_2N+kM'$) = 1/($k_2[B]$), for kM'<<$k_1[B]$; Also G = k'M'; and for kT<<k'R, i.e., the first term of Eq (14) for type-I, FRP is neglected. We obtain the following analytic solutions.

For case (i) unimolecular dominant, R = H/G, time integral of Eq (40) gives the CE' (for FRP) = 1- $M'/M'_0$, given by

$$CE(FRP) = 2k_2(bIA_0)t/M_0' \tag{43}$$

which is a linear increasing function of time (t), for the case of N = 0. Moreover, the CE for CP, given by Eq (15) is half of CE (FRP), that is CE(CP = 0.5 CE(FRP), for the case of N = 0.

For case (ii) bimolecular dominant, $R = [H/k']^{0.5}$, time integral of Eq (40) gives the CE for FRP as follows.

$$CE(FRP) = (1 - exp[-H'(t)])/M_0'$$ (44)

$$H'(t) = 2k'\sqrt{(2/(3k')bIA_0}\ t^{1.5}$$ (45)

The CE(CP) has the same formula as CE(FRP), but with 0.5H'(t), in Eq (44). We note that Eq (44) is a highly nonlinear function of time, comparing to the a linear increasing function of Eq (44).

We also that above solutions are based on $g = 1/(k_5 + k_2[B] + k_2N + kM') = 1/(k_2[B])$. We might have the condition that $g = 1/k_5$, then $[B] = B_0\ exp(-dt)$, with $d = (k_2/k_5)IbA_0$. In this case, the CE becomes. Eq (18) becomes

$$CE(FRP) = 2[1 - exp(-dt)]$$ (46)

and CE(CP) = 0.5 CE(FRP). Similarly, Eq (19) becomes,

$$CE(FRP) = 1 - exp[-H'(t)]$$ (47)

$$H'(t) = \left(4k'/\sqrt{dB_0}\right)[1 - exp(-dt)]$$ (48)

which is consistent with our previous formulas [18, 19], having the special feature that higher light intensity has a lower steady state value than that of lower intensity. These feature, based on $g = 1/k_5$, does not exist in Eq (44), or Eqs (46) and (47) using $g = 1/(k_2[B])$. The CE(FRP) in Eq (46) gives the CE(CP) with H'(t) reduced to 0.5H'(t).

Case (C) for G2/Iod/EDB system. We need to solve for [B] and N from Eqs (36) and (37) first. We will focus on the case that $g = 1/k_5$. The first-order solution, with $K_{12}[B] = 0$ in Eq (36) gives $[B] = B_0\ exp(-dt)$, which is used to solve for Eq (12), for the strong bimolecular case, with $R = [H/k']^{0.5}$, we obtain $N(t) = N_0\ exp[-(Q+Q')]$, $Q = (k_6/k')dB_0H'(t)$, $Q' = K_{12}B_0H'(t)$, with $H'(t) = [1-exp(-dt)]/d$. Solving for Eq (40), and for $H'(t) = dt$, $N(t) = N_0\ exp[-(Q'')t]$, with $Q'' = Q+Q'$, we obtain the CE for FRP as

$$CE(FRP) = 1 - exp[-P(t)]$$ (49)

$$P(t) = H'(t)[1 + HOR]$$ (50)

where the high-order term HOR is a complex function proportional to the time integral of $(K_{12}[B] + k_6R)N$, which needs numerical integration, having steady state value proportional to $[K_{12}B_0 + k_6(A_0/Ib)^{0.5}]N_0$. Solving for Eq (42), we obtain the CE for CP given by Eq (49), but replacing P(t) to 0.5 P(t).

The synergy effects (for improved CE) in an interpenetrated polymer network (IPN) system were discussed by Mau et al [17] for the polymerization of a TMPTA/EPOX blend in the presence of G2/Iod/EDB. Three factors were proposed as follows.

1. The FRP is at first inhibited by the oxygen in the medium, the OIH effects. However, while the cationic polymerization (CP) starts immediately which increases the medium viscosity limiting the diffusional oxygen replenishment, such that OIH is reduced;

2. The cationic monomer also acts as a diluting agent for the radical polymer network allowing to achieve a higher conversion.

3. the exothermic property of the radical polymerization also tends to boost the cationic polymerization that is quite temperature sensitive. As shown by Eqs (12) and (13), the radical (R) is reduced by the OIH term $k_5[O_2]$ in G. This OIH effects also suppress the CE of FRP, specially for the transient profile (till the oxygen is completely depleted). We note that this OIH does not affect the CE of CP, given by the radical S, which is independent to oxygen. This theoretical prediction was justified by the measured work of Mokbel et al [16], they reported that the radical for CP is much less sensitive to OIH than that of FRP. Mokbel et al [17] showed the OIH for system in air and in laminate, showing that the FRP conversion of TMPTA was higher in laminate than in air. In contrast, the CP conversion of epoxy function was lower in laminate than in air. It may be because the FRP of TMPTA was faster than the CP, and most of the free radicals were consumed to initiate FRP. A more general discussions for the synergic effects are shown in our recent Review article, Lin et al [18].

### 3.4. Analysis of measured data (G2 system)

1. Mau et al [17] reported that, in their Table 2 for G2/Iod/EDB system with TMPTA/EPOX monomer, the impact of the 10-fold reduction of copper complex (G2) initial concentration (from 0.7% to 0.07%) is particularly high on the CE of cationic polymerization (CE reduced from 51% to 30%). In contrast, it has much less impacting the free radical polymerization (CE reduced from 86% to 82%). This unique feature confirms our hypothesis on the IPN reactivity as photocatalyst in a photoredox cycle like G1. Mathematically, this could be realized by the REG term for regeneration of the initiator ([A] or, G2), in our Eq (1), in which for the strong RGE case, bI[A] = RGE, such that [A] is kept as a constant via the continuing regeneration of $A_0$.

2. The more sensitive dependence of CP than that of FRP on the G2 concentration ($A_0$) may be realized by comparing Eqs (9) and (10) as follows. Eq (9) for FRP is attributed from 3 coupling terms: the type-I T and M' coupling, the R and M' coupling and the T and N coupling. The $K_{12}[B]N$ term, shown in Eq (16), is for the extra radical R from Iod/EDB (without the light) and only attributes to CE of FRP. Eq (18) for the production of cation N shows that the CE of CP is also strongly dependent on $A_0$. In comparison, for the CE of FRP, Eq (16) has a rather high efficacy from the $K_{12}[B]N$ term, as shown by Fig. 5A of Mau et al [17], and it is not affected by $A_0$. Therefore, the impact of $A_0$ on the $k_2bIg\{A\}$ term of FRP is not as strong as that of CP.

3. The saturated CE in FRP is les sensitive to $A_0$. We have previously theorized the optimal concentration for maximum efficacy [19]. Moreover, we note that the catalytic cycle from the regeneration of the initiator [A] should also play important role, such that a small amount of initial G2 concentration (0.07%) is capable of being recycled for high efficacy, specially for FRP. However, the theoretically predicted optimal concentrations [19] for G2, Iod and EDM remain to be explored experimentally. The quantitative CE profiles can be produced and compared with that of Mau et al [17] using the solutions of Eqs (43) to (50) for various limiting cases, if the rate constants (kj, Kij) and other parameters, such as effective absorption constant (b), light intensity ($I_0$), and the initial concentrations of each co-initiators ($A_0$, $B_0$, $N_0$) are given.

### 3.5. General features and new findings (for G2 system)

As shown by the approximated solution of Eqs (44) to (50), the following significant features of the system G2/Iod/EDB [17] in monomer blend of TMPTA/EPOX are summarized.

1. Under the steady state condition, bI[A] = RGE in Eq (1), and d[A]/dt = 0, which leads to a prefect catalytic cycle such that [A] is a constant., [A] = $A_0$, We note that the REG enhances the conversion of FRP and CP, serving as a catalytic cycle. Without RGE (or a non-perfect RGE), [A] is depleted during the photopolymerization, given by a format of [A] = $A_0$Exp(-dt), leading to a lower steady state efficacy for both FRP and CP, unless there is a continuing supply of the initiator, such as in the clinical protocol of corneal cross-linking procedure, where more riboflavin solution drops are added during the procedure [19].

2. Both radicals of R (or $Ar^o$) and S' (or $EDB^o$) lead to FRP (in monomer TMPTA), and has a higher conversion efficacy than that of CP (in monomer EPOX), which is produced by only one radical S (or EDB(+)). As shown by Eqs (9), (10), (38) and (41), the rate function of FRP is about twice of CP, when N = 0. In the presence of N (or EDB), efficacy of FRP and CP increases due to the coupling terms of $K_6RN$ in Eqs (17) and (18) producing radical S' (or FRP) and cation S' for CP. The quantitative CE profiles can be produced and compared with that of Mau et al [17] using the solutions of Eqs (43) to (50) for various limiting cases, if the rate constants (kj, Kij) and other parameters are given.

3. Co-additive [B] has multiple functions of: (i) regeneration of initiator [A] leading to higher FRP and CP conversion; (ii) producing of radical S'(for FRP), and (iii) producing radical S (for CP), via the re-coupling with radical [C], produced by[B].

4. Co-additive [N] has functions of: (i) generation of S' for FRP; and (ii) generation of cation S for CP conversion; via $k_6R[N]$. Our analytic formulas show that CE (of CP) is bout 0.5 of CE(of FRP), as shown by Eqs (38) and (39). We note that the monomers used in the experimental data, TMPTA and EPOX, does not have the same functionality (TMPTA is a tri-functional monomer while EPOX is a difunctional monomer). Therefore, FRP and CP usually have not the same rate of polymerization.

5. The oxygen inhibition (OIH) effect (or the term $k''[O_2]$), reduces the free radicals R and S', and thus the FRP conversion, which is lower in air comparing to in laminate [22]. However, the OIH effect has much less impact on the CE of CP, as also demonstrated by Mau et al [17] experimentally, and shown by our formulas, Eqs (26) and (31).

6. In the IPN system, the synergic effects due to the co-exist of FRP and CP include: (i) CP can increase the medium viscosity limiting the diffusional oxygen replenishment, such that OIH is reduced; (ii) the cationic monomer also acts as a diluting agent for the radical polymer network, and (iii) the exothermic property of the radical polymerization also tends to boost the cationic polymerization that is quite temperature sensitive. We note that the overall efficacy of both FRP and CP are improved via the above described synergic effects, and most importantly, via the catalytic cycle from the regeneration of the initiator [A], the RGE term in Eq (1) which leads to a constant [A], when RGE = bI[A], such that d[A]/dt = 0. The present model, for the first time, confirmed mathematically this important feature, which was hypothesized by experimentalists [17].

7. We note that the catalytic cycle from the regeneration of the initiator [A] should also play important role in the dependence of efficacy on the copper complex (G2) concentrations, such that a small amount of initial G2 (0.07%) is capable of being recycled for high efficacy, specially for FRP, as hypothesized experimentally [18]. However, the theoretically predicted optimal concentrations [19] for G2, Iod and EDM remain to be explored experimentally.

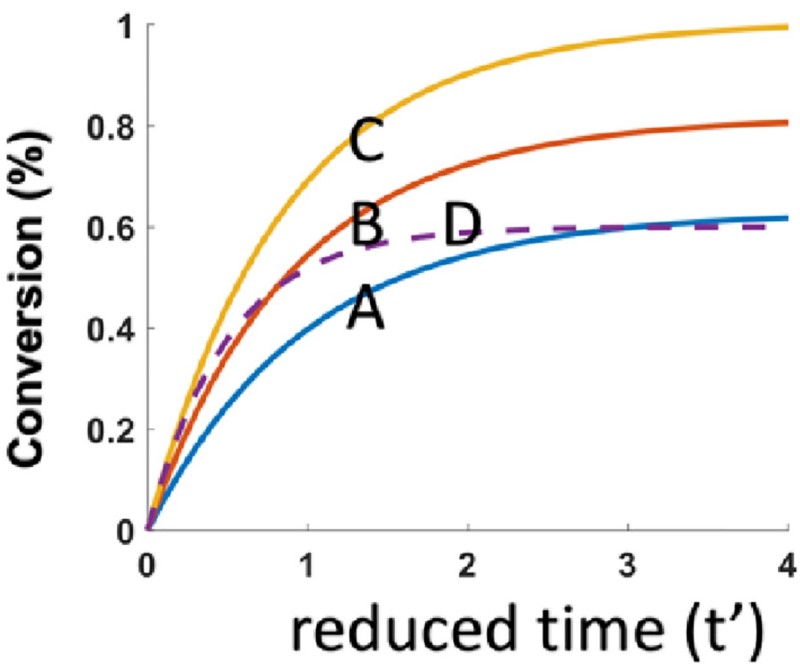

**Fig 1. Efficacy of FRP and CP for various $N_0$ = (0, 0.5, 1.0), for Curve-A, B, C (for FRP); and Curve-D (for CP0; for fixed b = 1.0 and $B_0$ = 0.6.**

## 3.6 Examples of numerical data

To demonstrate the above discussed features, we use Eq (30) for the CE of FRP and (31) for the CE of CP. For comprehensive results, we will use simplified and scaled/reduced parameters for the rate constants (which are not yet available) and focus on the role of the coinitiator initial concentration ($B_0$. $N_0$), and the absorption coefficients (b). Eqs (30) and (31) are simplified/reduced as follows:

$$CE' = [P(t') + N_0 P'(t')] \sqrt{b'B_0} \qquad (51)$$

$$CE = b'B_0 [1 - exp(-2b't')] \qquad (52)$$

with P(t) = [1-exp-bt)]; P'(t) = [1-exp(-1.5bt)], and t' is the reduced time such that dt = bt', and b' is a reduced coupling constant such that b' = $k_2$T'/(d$M_0$).

As shown by Fig 1, FRP (Curve-A,B,C) is always more efficient than CP (Curve-D), and FRP is an increasing function of the initial concentration $N_0$, or the enhanced efficacy due to coinitiator N. Fig 2 shows that the CE of both FRP and CP are increasing function of the initial concentration $B_0$. Fig 3 shows that the CE of FRP and CP are increasing function of absorption coefficients b. These results may be compared to the Fig. 9 of Mokbel et al [16], in which the 2 component system of G1/Iod, in their curve-1 is related to our case of $N_0$ = 0, having a lowest efficacy; and the system of BAPO/Iod/NVK, shown by curve-2 is less efficient that G1/Iod/NVK, curve-3 an dcurve-4 due to the lower b-value of BAPO than that of G1. Similarly. Fig. 15 of Mau et al [17] can be compared to our Fig 3 showing the role of b and the enhanced efficacy due to $N_0$, our Fig 1.

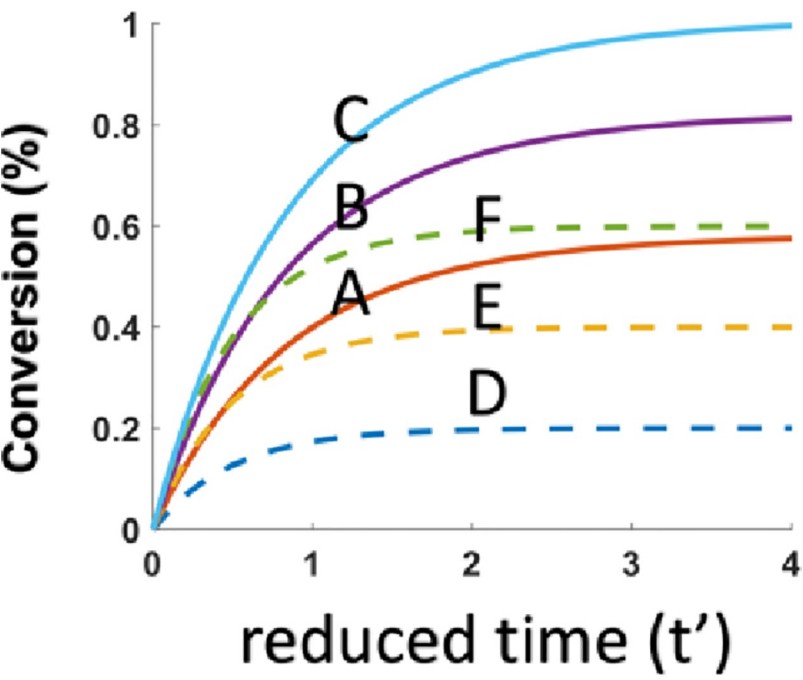

**Fig 2. Efficacy of FRP and CP for various $B_0$ = (0.2, 0.4, 0.6), for Curve-A, B, C (for FRP); and Curve-D, E, F (for CP);for fixed b = 1.0 and $N_0$ = 1.0.**

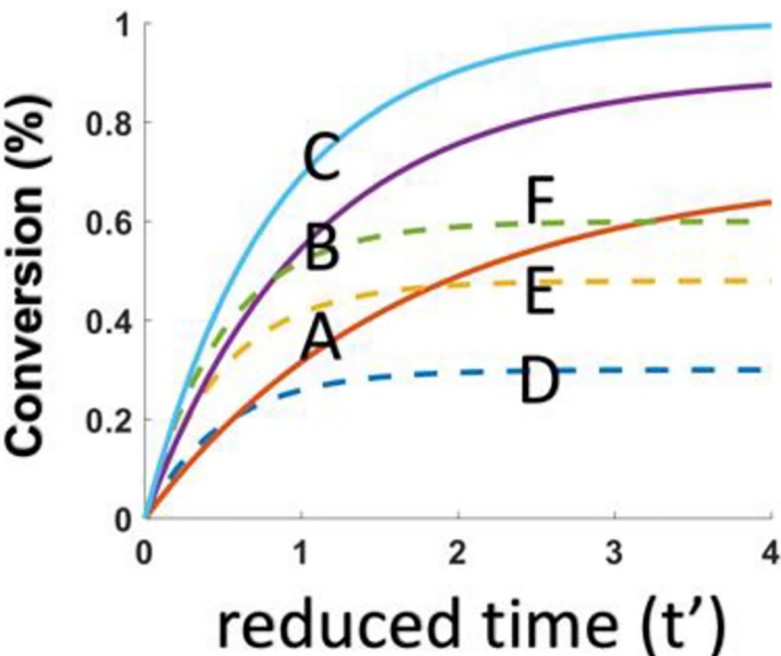

**Fig 3. Efficacy of FRP and CP for various absorption coefficients b = (0, 0.5, 1.0), for fixed $N_0$ = 1.0 and $B_0$ = 0.6.**

## 3.7. Further discussions

We note that Scheme 2 of the present article is more general than Scheme 3 based on Mau et al [17], which has ignored the bimolecular coupling term, as shown in Eq (16) of our kinetic equations. Furthermore, our general 3-component system, A/B/N, could include other components than the specific molecular groups of Mau at al [17], which is just the special case of our Scheme 2. For example, Scheme 1 could be slightly revised for another G1 system of Mokbel et al [16]. A more general discussions for the synergic effects are shown in our recent Review article, Lin et al [21].

There are several advantages offered by the cationic mode over free radical photoinitiated polymerization [2]. FRP is limited to monomers with olefinic double bonds, whereas compounds containing epoxide or vinyl ether groups can be polymerized by cation. However, many monomers that are prone to CP exhibit low volatility and negligible toxicity and possess good rheological properties. In contrast to FRP, molecular oxygen does not inhibit CP such that thick films can be cured in the presence of dry air. It is also well-known that water vapor terminates cationic polymerization. Furthermore, the problem of shrinkage in FRP of acrylic formulations negatively affects applications that require accurate part shape and size. However, the problems associated with volume shrinkage are less pronounced in CP, particularly when epoxy-based formulations are used and the polymerization is via a ring-opening process. Therefore, besides FRP, CP is also the method of choice in various applications. As reported by Mau et al [17], several characterisations of the acrylate/epoxy IPNs final properties have shown their advantages compared to pure acrylate or epoxy polymers. For example, the mechanical properties, adhesion properties, shrinkage and swelling can be improved but still more remarkably, the possibility to adapt or tune the IPN properties compared to radical or cationic polymerisations remains a unique advantage."

More features of copper complex catalyzed FRP and CP may be found in Refs [12, 22–27]. As a final remark, we note that the present article focuses on the free-radical-mediated FRP and cationic-catalyzed CP, in which available experimental results and proposed schemes of Mau et al [17] are used as the basis of our kinetic modeling. Other processes involving 3D (and 4D) printings shall also include the reversible deactivation radical polymerization (RDRP) techniques such as nitroxide-mediated polymerization (NMP) [30], atom transfer radical polymerization (ATRP) [31, 32], and reversible addition–fragmentation chain transfer (RAFT) [32]. However, they are not the scope of the present article, and they can be found in recent review articles by Corrigan et al. [33] and Bagheri et al. [34].

## 4. Conclusion

This article presents, for the first time, the kinetics and the general conversion features of the interpenetrated polymer network (IPN) capable of producing both FRP and CP in a blend of TMPTA and EPOX, as the monomers of FRP and CP, respectively. The synergic effects due to CP include: (i) the increased viscosity (via CP) limiting the diffusional oxygen replenishment, such that OIH are reduced; (ii) the cation also acts as a diluting agent for the IPN network, and (iii) the exothermic property of the CP. The new findings based on our formulas include: (i) the CE of FRP is about twice of the CE of CP, due to the extra radicals involved in FRP; (ii) the catalytic cycle enhancing the efficacy is mainly due to the regeneration of the initiator, the RGE term in Eq (1) which leads to a constant under the steady-state conditions, such that RGE = bI[A], or d[A]/dt = 0; (iii) the nonlinear dependence of light intensity of the CE (in both FRP and CP). For the first time, the catalytic cycle, synergic effects, and the oxygen inhibition are theoretically confirmed to support the experimental hypothesis [18]. The measured

results of Mokbel et al [16] and Mau et al [17] are well analyzed and matching the predicted features of our modeling.

## Author Contributions

**Conceptualization:** Jui-Teng Lin, Jacques Lalevee.

**Data curation:** Da-Chuan Cheng.

**Formal analysis:** Jui-Teng Lin.

**Funding acquisition:** Da-Chuan Cheng.

**Investigation:** Da-Chuan Cheng.

**Methodology:** Jui-Teng Lin.

**Project administration:** Da-Chuan Cheng.

**Resources:** Jacques Lalevee, Da-Chuan Cheng.

**Software:** Jui-Teng Lin.

**Supervision:** Da-Chuan Cheng.

**Validation:** Jacques Lalevee, Da-Chuan Cheng.

**Visualization:** Jacques Lalevee, Da-Chuan Cheng.

**Writing – original draft:** Jui-Teng Lin.

**Writing – review & editing:** Jui-Teng Lin.

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
