## [Decision Letter · Decision Letter 0]

21 Apr 2022

PONE-D-22-06318Efficacy analysis of new copper complex for visible light (455, 530 nm) radical/cationic photopolymerization: the synergic effects and catalytic cyclePLOS ONE

Dear Dr. Lin,

Thank you for submitting your manuscript to PLOS ONE. After careful consideration, we feel that it has merit but does not fully meet PLOS ONE’s publication criteria as it currently stands. Therefore, we invite you to submit a revised version of the manuscript that addresses the points raised during the review process.

We look forward to receiving your revised manuscript.

Kind regards,

Robert Chapman, Ph.D.

Academic Editor

PLOS ONE

Journal Requirements:

- https://pubmed.ncbi.nlm.nih.gov/34301082/

- http://turroserver.chem.columbia.edu/PDF_db/publications_901_950/NJT915.pdf

In your revision ensure you cite all your sources (including your own works), and quote or rephrase any duplicated text outside the methods section. Further consideration is dependent on these concerns being addressed.

"JTL thanks the internal grant of Medical Photon Inc. and DCC thanks the financial support from 359 China Medical University with the grant number CMU110-ASIA-11."

6. Thank you for stating the following in the Funding Section of your manuscript: 

"The Agence Nationale de la Recherche (ANR agency) is acknowledged for its financial support through the NoPerox grant."

We note that you have provided funding information that is not currently declared in your Funding Statement. However, funding information should not appear in the Funding section or other areas of your manuscript. We will only publish funding information present in the Funding Statement section of the online submission form. 

Additional Editor Comments: 

I accept the argument that the newly combined manuscript does present original research (contrary to the opinion of reviewer 1). However, based on the reviewers and my own assessment of this manuscript major revision is needed in two respects: a) the readability needs to be improved to better explain how the equations have been derived and; b) the correlation between these equations and published data needs to be better shown to demonstrate the validity of these results. Several minor comments have also been raised by the review process which should be addressed.

Reviewers' comments:

Reviewer's Responses to Questions

**Comments to the Author**

1. Is the manuscript technically sound, and do the data support the conclusions?

Reviewer #1: Partly

Reviewer #2: Partly

2. Has the statistical analysis been performed appropriately and rigorously? 

Reviewer #1: I Don't Know

Reviewer #2: N/A

3. Have the authors made all data underlying the findings in their manuscript fully available?

Reviewer #1: Yes

Reviewer #2: Yes

4. Is the manuscript presented in an intelligible fashion and written in standard English?

Reviewer #1: No

Reviewer #2: No

5. Review Comments to the Author

Reviewer #1: The authors aim to complement already published works from colleagues (Mokbel and al, Mau and al.) by developing analytic formulas for key parameters influencing the formation of IPNs by photopolymerization.

While the study appears to have clear objectives, the presented works lacks of novelty and originality compared to already published works (references 16, 17 and 22). In some degree, this submission seems more like a compilation of already known concepts than the presentation of original research.

Moreover, the language is unclear, making it difficult to follow (several typos and incomplete sentences). The authors should improve the readability of the text.

In my opinion, this topic has potential but this work need drastic revision:

- section 2.1 and 2.2 should be clarify for a better understanding of the developed formulas

- more correlation between the developed formulas and the published experimental data would be a great addition in section 3.1 and 3.2 in order to prove the trustworthiness of the formulas and their usefulness for simulation.

-L99, 156, 172 & 183: The authors refer to the active species for cationic polymerization (either "S" or "EDB(+)") as a radical, are you sure that it is not a cation radical?

-L205: The reaction r6 in scheme 4 seems unbalanced

-L218: What phenomenon is represented by the rate constant K12? Please add more explanation.

-L251: The authors claim the conversion of the cationic polymerization is reduced due to oxygen inhibition. However, cationic polymerization is reported in the literature, as not sensitive to oxygen. Please, justify your claim.

-L272: The authors refer to the concentration of [A] in Eq 16, please clarify.

-L374: Where does this formula come from. Please explain or add a reference.

-L505: Why have the authors consider the case of Iod/EDB without light? Please explain.

-L627: The authors have linked the difference in conversion between the free radical polymerization and the cationic polymerization to the number of species capable to initiate the reactions (2 for FRP, 1 for CP). However, the monomers used in the experimental data, TMPTA and EPOX, does not have the same functionality (TMPTA is a trifunctional monomer while EPOX is a difunctional monomer). Moreover, FRP and CP usually have not the same rate of polymerization. Please, include these facts in your explanation.

Reviewer #2: The manuscript describes kinetic schemes for two 3-initiator photochemical polymerization systems. The kinetics are complex, extending previous work of the authors and others. Some limiting cases within steady-state assumptions are examined analytically and the trends are compared with experimental results from previous work. I have no quarrel with the results or the analysis, except that it is not quantitative as claimed; however, the manuscript is difficult to read and I think it could be improved greatly, as detailed below.

1) For both G1 and G2, while the claim is made for a quantitative agreement with experiment, in fact only qualitative features are compared, increases or decreases etc. Reference is made to various figures from Refs. 16 and 17 and the reader is expected to follow the described trends mentally, imagining the figures. I would strongly suggest that pertinent figures be adapted to show both the experimental and the authors' approximate analytical solutions so that the reader can actually compare them.

2) The manuscript is the result of the fusion of two previous manuscripts and many errors have been made in the process. These must be carefully checked and corrected. Just a few examples (line 169, Scheme 3 should be Scheme 4; In Scheme 3, it should be G2, both in the scheme and in the legend; line 184, (r6) of Scheme 2 should be (r6) of Scheme 4; line 194, Scheme 1 should be Scheme 2); line 272, constant [A] in Eq. (16) should refer to Eq. (20); line 300 (12) and (13) are for G2 but this section is on G1, etc, etc).

3) Throughout the manuscript, replace oxodized by oxidized.

4) line 242, line 274, I don't think these are "comprehensive formulas". They are simplified formulas for some limiting cases.

5) line 291, S and S should be S and S'

6) lines 294 to 400, just one example of qualitative feature being misconstrued as quantitative: " faster raising rate"

6. PLOS authors have the option to publish the peer review history of their article (what does this mean?). If published, this will include your full peer review and any attached files.

Reviewer #1: No

Reviewer #2: No

---

## [Author Response · Author response to Decision Letter 0]

17 May 2022

Reply to academic editor comments:

1. we fixed style to meet PLosone.

2. we added DOI for refs.

3. in Figs 1 and 2, (and in the text) we cited Ref. [18] for overlaping text. In fact, 

 this article was written before Ref. [18] (a review article) was published. we cited Ref [18] in many places.

4. see revised cover letter for Funding inform which was not acceptable in your system. pls do online submission form on our behalf. 

5. we move the funding inform from Acknowledgement.

6. see 5.

Additional comments:

a) the readability needs to be improved to better explain how the equations have been derived and; 

 Our Reply: more text was added (shown in red).

b) the correlation between these equations and published data needs to be better shown to demonstrate the validity of these results.

 Our Reply: more text was added (shown in red).

REPLY to Rev.#1, 

- section 2.1 and 2.2 should be clarify for a better understanding of the developed formulas

- more correlation between the developed formulas and the published experimental data would be a great addition in section 3.1 and 3.2 in order to prove the trustworthiness of the formulas and their usefulness for simulation. 



REPLY: detailed comparison of experimental curves and our modeling require a complex numerical solutions, which is out of the scope of the present article (which focuses on simplified but analytic formulas).

Moreover, many of the rate constants (kj, Kj) in our modeling are not yet available (measured), therefore, it is impossible to have pertinent figures for a direct comparing to the measured curved. If the related rate constants are given (measured) then our derived formulas will lead to numerical roles of the key parameters, which are quantitative (not qualitative). Our developed simplified formulas, however, provide important features explored experimentally.



-L99, 156, 172 & 183: The authors refer to the active species for cationic polymerization (either "S" or "EDB(+)") as a radical, are you sure that it is not a cation radical?

REPLY: yes, agreed and fixed. Many thanks.

-L205: The reaction r6 in scheme 4 seems unbalanced

REPLY: yes, we fixd it by EDBo(+)

-L218: What phenomenon is represented by the rate constant K12? Please add more explanation.

REPLY: we add these text: In Eq. (17) and (18), K12 is the coupling between Iod (B) and EDB (N), which also produces extra radical R (without the light), as shown by our r6. However the FRP due to this extra R is very small (for a week coupling constant of K12), as reported in Ref [17].

-L251: The authors claim the conversion of the cationic polymerization is reduced due to oxygen inhibition. However, cationic polymerization is reported in the literature, as not sensitive to oxygen. Please, justify your claim.

REPLY: agreed and fixed.

L272: The authors refer to the concentration of [A] in Eq 16, please clarify.

REPLY: corrected as Eq. (19).

-L374: (386) Where does this formula come from. Please explain or add a reference. REPLY: we added Ref. [21,22].

-L505: Why have the authors consider the case of Iod/EDB without light?Please explain.

 REPLY: the Iod/EDB was used as a standard to ensure that the EDB/Iod charge transfer complex could not initiate the polymerization, if lack of absorption for the charge transfer complex at a non-absorbing light wavelength (or no light). It was also measured in ref [17].

-L627: The authors have linked the difference in conversion between the free radical polymerization and the cationic polymerization to the number of species capable to initiate the reactions (2 for FRP, 1 for CP). However, the monomers used in the experimental data, TMPTA and EPOX, does not have the same functionality (TMPTA is a trifunctional monomer while EPOX is a difunctional monomer). Moreover, FRP and CP usually have not the same rate of polymerization. Please, include these facts in your explanation.

REPLY: yes, we added text in L 676-679. 

Reply to Rev.#2 ***********************************************************

1) For both G1 and G2, while the claim is made for a quantitative agreement with experiment, in fact only qualitative features are compared, increases or decreases etc. Reference is made to various figures from Refs. 16 and 17 and the reader is expected to follow the described trends mentally, imagining the figures. I would strongly suggest that pertinent figures be adapted to show both the experimental and the authors' approximate analytical solutions so that the reader can actually compare them. 



REPLY: a detailed comparison of experimental curves and our modeling require a complex numerical solutions, which is out of the scope of the present article (which focuses on simplified but analytic formulas).

Moreover, many of the rate constants (kj, Kj) in our modeling are not yet available (measured), therefore, it is impossible to have pertinent figures for a direct comparing to the measured curved. See more discussion in our REPLY after comments 9). 



2) The manuscript is the result of the fusion of two previous manuscripts and many errors have been made in the process. These must be carefully checked and corrected. 

Yes, we have switched the definition of G1 and G2, such that G1 is for the first-system and G2 is for the second system to avoid errors caused by them.

Just a few examples



(1) (line 169, Scheme 3 should be Scheme 4; In Scheme 3, it should be G2, both in the scheme and in the legend;

REPLY: fixed as G1, we have switched G2 and G1.

(2) line 184, (r6) of Scheme 2 should be (r6) of Scheme 4; fixed

(3) line 194, Scheme 1 should be Scheme 2); fixed

(4) line 272, constant [A] in Eq. (16) should refer to Eq. (20); fixed

(5) line 300 (12) and (13) are for G2 but this section is on G1, etc, etc).

REPLY: fixed. we have switched G1 and G2 for consistent.

6) Throughout the manuscript, replace oxodized by oxidized.

7) line 242, line 274, I don't think these are "comprehensive formulas". They are simplified formulas for some limiting cases. partially agreed and fixed

8) line 291, S and S should be S and S' fixed

9) lines 294 to 400, just one example of qualitative feature being misconstrued as quantitative: " faster raising rate".



REPLY: we partially agreed. In fact, the discussions shown in sections 3.2 (for G1 system) and 3.4 (for G2 system) are based on our analytic formulas which defined specifically and quantitatively the roles of each of the key parameters for the conversion efficacy. Our formulas predict many of the measured data (and their general trends), besides other to-be-explored features. If the related rate constants such as kj and Kij, are given (measured) then our derived formulas will lead to numerical roles of the key parameters, which are quantitative (not qualitative).

---

## [Editor Report · Decision Letter 1]

9 Jun 2022

PONE-D-22-06318R1Efficacy analysis of new copper complex for visible light (455, 530 nm) radical/cationic photopolymerization: the synergic effects and catalytic cyclePLOS ONE

Dear Dr. Lin,

Thank you for submitting your manuscript to PLOS ONE. After careful consideration, we feel that it has merit but does not fully meet PLOS ONE’s publication criteria as it currently stands. Please see my detailed comments below. Therefore, we invite you to submit a revised version of the manuscript that addresses the points raised during the review process.

We look forward to receiving your revised manuscript.

Kind regards,

Robert Chapman, Ph.D.

Academic Editor

PLOS ONE

Journal Requirements:

Additional Editor Comments (if provided):

The authors have responded to the first point (a) raised by the reviewers - the revised manuscript is more readable and the derivations better eplained. The derivations are detailed and accurate (as they were in the original submission). However, they have still not answered the main comment raised by both revieiwers - that there is no comparison of the results of the analytical solutions with experimental data.

Section 3.2 and 3.4 attempt to describe (in words) the similarities between the equations and the experimental data, which is helpful. They claim that fitting some predicted values of k, in order to plot quantitative solutions against the experimental data requires “complex numerical solutions” that are outside the scope of the present article, which focusses instead on simplified formulae.

It’s not clear why this should be the case. For example, there is only one unknown in eq. 31. The authors assert in line 416-421 that given the values of bI0 and the initial concentrations, they could calculate and compare the profiles of CP, but are hindered by lack of experimental ki and kij values. If this is true, then it should be no problem to demonstrate this by plotting the function with some fitted or assumed values of kj and kij to provide testable predictions for these and to demonstrate the validity of their equations. The same is true in points b, c, d, and e of this section and in section 3.4. Rather than just asserting the similarities between the equations and the experimental data it ought to be possible to demonstrate this by plotting the data, even without the experimentally derived values of the rate constants. Even very simple and approximate plots would permit the authors then to draw the conclusion that they have determined kinetic schemes that they have presented equations for the "kinetics and general conversion features of an IPN”.

Producing such figures (even in the absence of experimental data, to show the general trends align, or perhaps on just one set of input initiator values) should be a matter of hours and would completely put to rest this question? If this is not the case, can the authors please explain why not??

Minor typos to fix:

L133: “coupling” not “conpling”

L135: “example” not “examples"

L446: “similar to THE G1 system of 3.1”

L668: “Eqs.(“ missing text

---

## [Author Response · Author response to Decision Letter 1]

13 Jun 2022

REPLY to Comments;

Additional Editor Comments (if provided):

The authors have responded to the first point (a) raised by the reviewers - the revised manuscript is more readable and the derivations better eplained. The derivations are detailed and accurate (as they were in the original submission). However, they have still not answered the main comment raised by both reviewers - that there is no comparison of the results of the analytical solutions with experimental data.

Producing such figures (even in the absence of experimental data, to show the general trends align, or perhaps on just one set of input initiator values) should be a matter of hours and would completely put to rest this question? If this is not the case, can the authors please explain why not??

REPLY: yes, we agreed and we have added Section 3.5 to include figs 1 to 3 to show the roles of key parameters, which are related to figs (measured data) of prior articles.

Minor typos to fix: (all fixed in "red")

L133: “coupling” not “conpling”

L135: “example” not “examples"

L446: “similar to THE G1 system of 3.1”

L668: “Eqs.(“ missing text

---

## [Editor Report · Decision Letter 2]

15 Jun 2022

Efficacy analysis of new copper complex for visible light (455, 530 nm) radical/cationic photopolymerization: the synergic effects and catalytic cycle

PONE-D-22-06318R2

Dear Dr. Lin,

We’re pleased to inform you that your manuscript has been judged scientifically suitable for publication and will be formally accepted for publication once it meets all outstanding technical requirements.

Kind regards,

Robert Chapman, Ph.D.

Academic Editor

PLOS ONE

Additional Editor Comments (optional):

The authors have responded to all the comments of the reviewers - the new figures help show that the equations could be used to fit the experimental data.
---

## [Editor Report · Acceptance letter]

14 Jul 2022

PONE-D-22-06318R2 

Efficacy analysis of new copper complex for visible light (455, 530 nm) radical/cationic photopolymerization: the synergic effects and catalytic cycle 

Dear Dr. Lin:

I'm pleased to inform you that your manuscript has been deemed suitable for publication in PLOS ONE. Congratulations! Your manuscript is now with our production department. 

Kind regards, 

on behalf of

Dr. Robert Chapman 

Academic Editor

PLOS ONE